# NMR-Based Metabolic Profiles of Intact Zebrafish Embryos Exposed to Aflatoxin B1 Recapitulates Hepatotoxicity and Supports Possible Neurotoxicity

**DOI:** 10.3390/toxins11050258

**Published:** 2019-05-08

**Authors:** Zain Zuberi, Muhamed N. H. Eeza, Joerg Matysik, John P. Berry, A. Alia

**Affiliations:** 1The School of Pharmacy and Pharmaceutical Sciences, Trinity College, D02 PN40 Dublin, Ireland; zuberiz@tcd.ie; 2Leiden Institute of Chemistry, Leiden University, 2333 Leiden, The Netherlands; 3Institute for Medical Physics and Biophysics, University of Leipzig, 04107 Leipzig, Germany; MuhamedNour.HashemEeza@medizin.uni-leipzig.de; 4Institute for Analytical Chemistry, University of Leipzig, 04107 Leipzig, Germany; joerg.matysik@uni-leipzig.de; 5Department of Chemistry and Biochemistry, Florida International University, Miami, FL 33181, USA

**Keywords:** aflatoxin B1, zebrafish, high-resolution magic angle spin (HRMAS), nuclear magnetic resonance (NMR), metabolomics, biomarkers

## Abstract

Aflatoxin B1 (AFB1) is a widespread contaminant of grains and other agricultural crops and is globally associated with both acute toxicity and carcinogenicity. In the present study, we utilized nuclear magnetic resonance (NMR), and specifically high-resolution magic angle spin (HRMAS) NMR, coupled to the zebrafish (*Danio rerio*) embryo toxicological model, to characterize metabolic profiles associated with exposure to AFB1. Exposure to AFB1 was associated with dose-dependent acute toxicity (i.e., lethality) and developmental deformities at micromolar (≤ 2 µM) concentrations. Toxicity of AFB1 was stage-dependent and specifically consistent, in this regard, with a role of the liver and phase I enzyme (i.e., cytochrome P450) bioactivation. Metabolic profiles of intact zebrafish embryos exposed to AFB1 were, furthermore, largely consistent with hepatotoxicity previously reported in mammalian systems including metabolites associated with cytotoxicity (i.e., loss of cellular membrane integrity), glutathione-based detoxification, and multiple pathways associated with the liver including amino acid, lipid, and carbohydrate (i.e., energy) metabolism. Taken together, these metabolic alterations enabled the proposal of an integrated model of the hepatotoxicity of AFB1 in the zebrafish embryo system. Interestingly, changes in amino acid neurotransmitters (i.e., Gly, Glu, and GABA), as a key modulator of neural development, supports a role in recently-reported neurobehavioral and neurodevelopmental effects of AFB1 in the zebrafish embryo model. The present study reinforces not only toxicological pathways of AFB1 (i.e., hepatotoxicity, neurotoxicity), but also multiple metabolites as potential biomarkers of exposure and toxicity. More generally, this underscores the capacity of NMR-based approaches, when coupled to animal models, as a powerful toxicometabolomics tool.

## 1. Introduction

Aflatoxins are potent mycotoxins specifically produced by the fungal genus *Aspergillus*, which is endemic to soil organic matter, but also among the most common fungal contaminants of food and feed crops including, in particular, grains (e.g., rice, corn, wheat), nuts/seeds, legumes, and various other staples (for a current review, see [1]). Contamination of crop plants by *Aspergillus* and, in turn, aflatoxin can occur both pre-harvest (i.e., fungal infection of plants) and post-harvest (i.e., storage, spoilage). Aflatoxins are consequently detected in a wide range of plant and animal-based (e.g., milk, eggs, meat) agricultural products. In fact, aflatoxins were first discovered in the 1960s following an outbreak of so-called “turkey X disease”, which resulted in a massive (>100,000) number of deaths among domestic turkeys linked, in turn, to contamination of feed grains by *A. flavus*, as well as particularly high sensitivity of poultry to aflatoxins [2,3].

Given the recognized adverse effects of aflatoxins on human health, regulatory limits have been adopted by many nations. However, similar guidelines are often either not uniformly established or effectively enforced in many developing countries due, in particular, to widespread subsistence farming, which makes monitoring and regulation exceedingly difficult [4]. Exacerbating exposure risk in this context, conditions favorable for *Aspergillus* contamination of agricultural crops primarily occur within the tropical and sub-tropical climates (i.e., 40° N–40° S latitude), where economically-developing nations are disproportionately located. Accordingly, it has been estimated that as many as 4.5 billion people worldwide are chronically exposed to aflatoxins [5].

Toxicologically, aflatoxins are associated with both acute and sub-acute adverse health effects. With respect to acute intoxication, aflatoxicosis following exposure to high concentrations of aflatoxins, which primarily target hepatocytes, leads to hemorrhagic necrosis, edema, and other symptoms associated with liver damage (e.g., bile duct proliferation, jaundice, pain, vomiting) and can frequently (>30% of cases) be fatal, particularly among children [5,6,7]. Diagnosis of aflatoxicosis in humans, however, is difficult due to variability in clinical manifestations and because toxicity is often complicated by other co-contributors including infectious disease, immunosuppression, and nutrition. The potential toxicity of aflatoxin, alongside higher exposure risk, is thereby further exacerbated in developing countries, in particular due to the prevalence of these co-contributors (i.e., high rates of infectious and/or immunosuppressive diseases, malnutrition) in these regions. Consequently and despite decades of awareness regarding their potential to contaminate agricultural crops, fatal outbreaks of acute aflatoxicosis continue to be reported with some regularity among populations within, in particular, Sub-Saharan Africa and Asia [5]. While acute intoxications are relatively rare and largely limited geographically, aflatoxins are thought to contribute worldwide to sub-acute health impacts including, in particular, cancer.

Of the more than 20 variants produced, aflatoxin B1 (AFB1) is generally considered the most toxic variant. In addition to acute toxicity, AFB1 is classified by the International Agency for Research on Cancer (IARC) as a Group 1 carcinogen (i.e., “carcinogenic to humans”). The carcinogenic potential of AFB1 is specifically linked to hepatocellular carcinoma (HCC), which accounts for as much as 70% of human cancer deaths in areas where aflatoxicosis is endemic [8,9], and it has been estimated that as much as 28% of HCC worldwide may be attributed to AFB1 [10]. It is generally established that metabolic bioactivation of AFB1 is key to carcinogenicity and teratogenicity (via DNA alkylation), as well as acute toxicity [3]. Specifically, phase I metabolism of AFB1 in the liver by cytochrome P450 enzyme systems, and particularly CYP1A and CYP3A subfamilies, leads to the production of epoxides including, in particular, AFB1 exo-8,9-epoxide (AFBO). The epoxide is, in turn, considered to be the primary mutagenic metabolite via the formation of guanine-based DNA adducts [1]. Alongside carcinogenicity, AFB1 has been additionally linked to other sub-acute health effects including interference with micronutrient adsorption (and consequent malnutrition), growth impairment, and immunosuppression [1,5]. Yet, despite established links between AFB1 and both acute and various sub-acute health impacts, pathways of toxicity remain to be fully elucidated.

In the present study, we utilized techniques based on nuclear magnetic resonance (NMR), specifically applied to the zebrafish (*Danio rerio*) embryo model, to investigate metabolic changes associated with AFB1 exposure. Embryonic and other early life (e.g., larval) stages of the zebrafish are well established as a toxicological model, in general, and have been specifically shown in several previous studies [11,12,13] to be an effective model for assessment of AFB1 toxicity. It is worth noting that, in addition to contamination of agricultural products, AFB1 is one of the most common contaminants of aquaculture (i.e., fish) feeds [14]. As such, assessment of the toxin in the zebrafish system not only represents a model for human and mammalian toxicity, but may also have direct relevance to the field of aquaculture.

Herein, we couple the zebrafish as a model system to high-resolution magic-angle spinning (HRMAS) NMR techniques, which has been recently shown to enable both highly quantitative and qualitative (i.e., metabolite identification) analyses of major metabolites in the developing zebrafish embryo [15,16,17,18,19]. Application of this metabolomics approach to the zebrafish embryo model has, in fact, been previously demonstrated with respect to other naturally-occurring toxicants and in these prior studies shown to facilitate both characterization of toxicological pathways and identification of possible biomarkers of toxin exposure [18,19]. The present study exploits the power of this technique to both characterize toxicological pathways toward better understanding of how AFB1 toxicity translates to adverse health outcomes and identify potential metabolic biomarkers of exposure (and toxicity) toward improved means of tracking exposure and human health impacts.

## 2. Results

### 2.1. Toxicity of AFB1 in the Zebrafish Embryo Model

Acute toxicity of AFB1, based on embryo lethality, was evaluated over a range of concentrations (up to 2 µM) in zebrafish embryos, specifically following 24 h of exposure at representative developmental stages (i.e., 24, 48, 72, and 96 hpf). Dose-dependent toxicity was observed (at all exposure stages) below 2 µM; moreover, toxicity was clearly stage-dependent and specifically increased with age of embryos (Figure 1). AFB1 was, for example, nearly an order of magnitude more toxic for embryos exposed at 96 hpf compared to 24 hpf, and there was, more generally, a stage-dependent decrease in calculated LC_50_ (after 24 h exposure) of 2.1, 1.8, 1.1, and 0.5 µM at 24, 48, 72, and 96 hpf, respectively. 

At sub-lethal concentrations (below LC_50_), AFB1 impaired development, resulting in embryo deformity. Deformities were generally observed among approximately 20–30% of surviving embryos at or below lethal concentrations. At 72 hpf, for example, embryos exposed to concentrations below the approximate LC_50_ (i.e., 1 µM) were consistently characterized by malformation of the head and bending of the tail and upper body (Figure 2). 

### 2.2. NMR-Based Metabolic Profiles of Zebrafish Exposed to AFB1

High-resolution magic angle spin NMR resolved several metabolites in intact zebrafish embryos (Figure 3) and when coupled to principal components analysis (PCA) enabled statistical discrimination (Appendix A) of quantitative differences in metabolites between AFB1-exposed and control (i.e., DMSO only) embryos. Comparisons (based on 1-D NMR chemical shifts) to the Human Metabolome Database (HMDB), along with 2D NMR techniques (i.e., ^1^H-^1^H COSY) enabled unambiguous identification, and subsequent quantitation, of metabolites (Figure 4). Consequently, 28 metabolites could be identified, and consequently quantified and statistically evaluated. 

Of these, a total of 19 metabolites were shown to increase or decrease significantly (*p* < 0.05) following 24-hour exposure to AFB1 at 72 hpf (Figure 4 and Table 1). A significant increase of several amino acids including phenylalanine (Phe, *p* < 0.01), tryptophan (Trp, *p* < 0.001), and tyrosine (Tyr, *p* < 0.0001), as well as isoleucine (Ile, *p* < 0.05), glutamate (Glu, *p* < 0.05), glutamine (Gln, *p* < 0.05), and glycine (Gly, *p* < 0.05) was observed, whereas a highly significant (*p* < 0.0001) decrease in cysteine (Cys) was measured. Chemical shifts used to identify amino acids are specific for amino acids that are not incorporated into protein, and quantification, therefore, reflects the concentration of the “free” amino acid pools for each. Notably, the non-proteinogenic amino acid neurotransmitter, γ-aminobutyric acid (GABA), also significant increased (*p* < 0.05). Numerous metabolites associated with carbohydrate metabolism, and cellular energetics, were significantly altered by AFB1 treatment including: (1) decreases in glucose-1-phosphate (G1P) and glucose-6-phosphate (G6P), as well as glucose (Glc; *p* < 0.001) itself; (2) highly significant increases in lactate (Lac, *p* < 0.0001) as the product of lactate dehydrogenase and/or anaerobic glycolysis; and (3) increases in several metabolites associated with cellular energetics including ATP, NADH, and NAD^+^ (*p* < 0.05). Significant increases in fatty acids (FA, *p* < 0.001) and cholesterol (Chol, *p* < 0.01) were observed, alongside a concomitant increase in acetate as both an intermediate of lipid metabolism and anaplerotic catabolism (i.e., β-oxidation) into the Krebs cycle (as acetyl CoA). Alongside changes in lipids, significant increases in the polar headgroups, choline (Cho) and *myo*-inositol (m-Ins), of phospholipids’ characteristic cellular membranes were observed. Finally, a significant decrease (*p* < 0.05) in glutathione (GSH) as a phase II detoxification mechanism was measured following AFB1 exposure (compared to controls). 

## 3. Discussion

Contamination of crop plants by aflatoxinogenic *Aspergillus* has been clearly linked to both acute intoxication (i.e., “aflatoxicosis”) and carcinogenicity, although a complete picture of the pathways of toxicity remains to be fully clarified. To elucidate pathways and potential exposure biomarkers, we utilized HRMAS NMR to characterize alterations in the metabolic profiles of intact zebrafish embryos following exposure to AFB1. Solution-state NMR techniques have been previously applied ex vivo [20,21,22,23] to metabolomics studies of AFB1 in mammalian systems; however, the present study represents the first to utilize HRMAS NMR of an intact organismal model to understand the toxicology of AFB1. This approach in the zebrafish system has, indeed, been previously demonstrated and shown to be highly effective, with respect to other environmental toxicants including, in particular, aquatic (i.e., algal) biotoxins [18,19]. Similar to these prior studies, when coupled to toxicological assessment, this approach enabled the development of a holistic and integrated model of AFB1. 

### 3.1. Toxicity of AFB1 in the Zebrafish Embryo Model

Aligned with previous studies [11,12,13], ambient exposure to AFB1 was lethal to zebrafish embryos in the micromolar range (Figure 1). Developmental effects (Figure 2), within this same exposure range, were both quantitatively and qualitatively consistent with these previous studies. Prior reports of the embryotoxicity of AFB1 in the zebrafish model [11] determined comparable lethal concentrations (LC_50_ = 2.3 µM, versus 1.1 µM in the present study; see Figure 1) for 72 hpf embryos exposed to AFB1 for 24 h and observed similar developmental deformities including, in particular, deformity of the head, tail, and body axis (Figure 2). Micromolar exposure concentrations associated with lethality, and other various developmental endpoints, have been similarly confirmed in the zebrafish embryo model by more recent studies [12], and the time dependence (>72–96 hpf) of zebrafish embryotoxicity at sub-micromolar concentration, as observed in the present study (Figure 1), has been, likewise, very recently reported [13]. Notably, however, in the present study, toxicity was specifically observed with a sequential exposure regime (i.e., 24 h at 24, 48, 72, and 96 hpf; Figure 1), rather than continuous exposure (as in all prior studies), clearly indicating stage-dependent susceptibility (as opposed to possible cumulative toxicity). The observed concentration- and stage-dependent toxicity was, in turn, referenced to select appropriate exposure concentrations and stages for NMR analyses (see below). Specifically, these subsequent studies utilized 72 hpf embryos exposed (for 24 h) to a concentration of 1 µM, which approximates the LC_50_ of AFB1 (with additional replicates to provide a sufficient number of embryos; see Materials and Methods).

Given the reported role of hepatocytes and, moreover, cytochrome P450 enzymes (which are primarily localized to the liver) in the metabolic bioactivation AFB1 to reactive epoxides (e.g., AFBO), it is proposed that observed stage-dependence of toxicity in the zebrafish embryo is related to the development of the liver and associated expression of these phase I detoxification enzymes. Although initial differentiation (i.e., “budding”) of the liver in zebrafish embryos begins at 24 hpf, complete development (i.e. “outgrowth and expansion”) does not occur until approximately 72–96 hpf [24]. Moreover, significant expression of relevant cytochrome P450 enzymes (e.g., CYP1A and CYP3A), and corresponding xenobiotic-induced activity, in the zebrafish embryo does not occur until approximately 72 hpf and is primarily localized to the developing liver [25,26]. Relevant to the present results (see below) also, phase I metabolism by CYP enzymes is coupled, in turn, to subsequent phase II detoxification, and specifically conjugation of glutathione (GSH) to electrophilic AFBO by glutathione-S-transferase (GST) [27] (see Figure 5). Toxicity of AFB1 is, therefore, linked to levels of GSH and GST activity. It has been shown, for example, that elevated sensitivity of certain poultry (e.g., domestic turkey and “turkey X disease”) to AFB1 is due to a combination of highly efficient CYP1A and CYP3A enzymes (and conversion to reactive epoxides) and simultaneously deficient GST activity, in the hepatocytes of these species [3].

### 3.2. Alteration of Metabolic Profiles of Zebrafish Embryos by AFB1: Development of a Toxicological Model

Metabolic profiling of zebrafish embryos by HRMAS NMR (Figure 4 and Table 1) in the present study is remarkably consistent with the hepatotoxicity of AFB1 and recapitulates many of the ex vivo observations in previous metabolomics studies in other (i.e., mammalian) models [20,21,22,23]. One of the key cellular aspects of hepatotoxicity is the swelling of cells (i.e., hepatocytes), which in turn, results in the release and subsequent hydrolysis of cell membrane components [20,28]. In the present study, HRMAS NMR accordingly measured significant increases in both lipids (i.e., FA and Chol) and the primary polar headgroups (i.e., m-Ins and Cho) associated with phospholipids that are essential to cell membranes (Figure 4). In addition, increased Lac, in our study, is consistent with elevated LDH activity that is widely recognized as a general measure of cytotoxicity and, in the current model, a possible indicator of hepatic cytotoxicity of AFB1 [29]. At the same time, the observed decrease in GSH aligns with the essential role of phase II hepatic detoxification of AFB1. Following phase I metabolic bioactivation of AFB1 to AFBO, conjugation of GSH via GST facilitates removal of these reactive species, and observed decreases in GSH would, therefore, be consistent with metabolic consumption of the glutathione pool in response to the formation of the reactive AFB1 epoxide (Figure 5). 

Alongside these general indicators of hepatotoxicity, in our study, the measured alterations of three major metabolic pathways—namely amino acid, lipid, and carbohydrate (i.e., energy) metabolism (Figure 4 and Table 1)—which are primarily localized to the liver, are, likewise, highly consistent with both hepatic targeting of AFB1 and previous studies in mammalian systems [20,21,22,23]. Based on observed alterations in metabolic profile, an integrated model of the toxicity of AFB1 in the zebrafish embryo model is proposed (Figure 5).

Central to these metabolic changes is a significant alteration of the levels of several relevant amino acids following exposure of zebrafish embryos to AFB1. Exposure of 72 hpf embryos to AFB1 altered levels of several amino acids including significant increases of Phe, Tyr, Trp, Ile, Glu, Gln, and Gly and a significant decrease of Cys (Figure 4 and Table 1). Altered amino acid levels have, indeed, been consistently identified in relation to hepatic pathology, in general, and hepatotoxicity of AFB1 specifically, in effectively all previous metabolomics studies [20,21,22,23]. As an established metabolic biomarker of liver damage, coined the “Fischer ratio”, it has, for example, long been known that during hepatic failure, the ratio of aromatic amino acids (AAA; i.e., Tyr, Phe, and Trp) to branched-chain amino acids (BCAA; i.e., Leu, Ile, and Val) increases [30,31,32]. This is related, in part, to the fact that catabolism of BCAA, in contrast to AAA (and, indeed, all other amino acids), is not localized to the liver, but rather peripheral systems, and particularly skeletal muscle [33]. The ratio of the normalized levels of AAA:BCAA and Tyr:BCAA (as a variant of the Fischer ratio [32]) for 96 hpf embryos significantly increased with exposure to AFB1 relative to controls (i.e., 1.5-fold and 1.8-fold, respectively; Table 1). Of the BCAA, a statistically-significant change (increase) of only Ile was observed; however, highly significant increases in all AAA (i.e., Tyr, Phe, and Trp) were measured for 72-hpf AFB1 exposures (Figure 4), which resulted in the elevated AAA:BCAA and Tyr:BCAA ratios (Table 1). The observed increase in Ile is very notable since this particular BCAA has been distinctively demonstrated in several studies [34,35,36,37,38] to increase glucose uptake and utilization and decrease gluconeogenesis by liver. The observed increase in Ile would, therefore, be consistent with increased utilization of glucose (from glycogen) observed here (Figure 5; see the discussion below). Both AAA and BCAA are known to be essential amino acids for teleost fish [39], such as zebrafish, and are necessarily derived from diet (or, in the case of embryos, protein-rich yolk) such that any change in their levels is presumptively due to alterations in their catabolism, rather than anabolism (i.e., biosynthesis). 

In addition, levels of the non-essential amino acids Glu, Gln, and Gly were, likewise, elevated in AFB1-exposed embryos. Elevated Glu is noteworthy, in this regard, as conversion of Glu to α-ketoglutarate (αKG) via either transamination or glutamate dehydrogenase represents a key alternative (to pyruvate/acetyl CoA) as an entry point for the Krebs cycle to meet cellular energy demands (Figure 5). Furthermore, “upstream” deamination of Gln via glutaminase, likewise, provides the substrate (i.e., Glu) for transamination to αKG (and subsequent entry to the Krebs cycle), and it has been asserted that so-called glutaminolysis (alongside glycolysis; see below) is essential to metabolic homeostasis during embryo development [40,41]. It is, therefore, proposed that cellular damage to hepatocytes, associated with AFB1 exposure, leads to loss of context-specific glutaminase and transaminase activity and, consequently, impaired amino acid catabolism (and homeostasis, more generally), which contributes to toxicity in zebrafish embryos. This notion is supported, for example, by previous studies of the hepatotoxicity of acetaminophen in the zebrafish embryo, whereby a similar stage- and concentration-dependent embryotoxicity is correlated with the elevation of serum levels, and consequent loss in hepatocytes, of transaminase activity [42]. 

Interestingly, neither Asp, nor Ala levels were significantly altered (Figure 4). This is notable given their well-known role in amino acid biosynthesis and catabolism, and specifically transamination reactions (including conversion of Glu to αKG) in the liver. Plasma levels of aspartate and alanine transaminases (i.e., AST and ALT, respectively) and, moreover, their ratio (i.e., AST/ALT) are some of the best-established biomarkers of liver damage (although other amino acid transaminases have been, likewise, linked to hepatic pathology) [43]. Specifically, increased plasma levels of AST and ALT are linked to damage to the liver, as the primary location of these enzymes, which leads to their release into plasma. Although ALT and AST were not directly measured in the present study, decreased transaminase capacity of hepatocytes (due to hepatocytotoxicity) might be expected to alter Ala and Asp (as substrates of these enzymes). However, although levels of both Asp and Ala were elevated, neither was increased significantly (*p* > 0.05; Figure 4 and Table 1). The lack of a significant change in Ala and Asp is proposed to be due, in part, to the lack of transamination by mitochondrial ALT and AST (i.e., *m*ALT and *m*AST) of Glu to αKG, to supply the Krebs cycle, for which these two amino acids are the primary products. Specifically, previously-demonstrated disruption of mitochondrial function including Krebs cycle and coupling of oxidative phosphorylation ([44], see below) would, in this proposed mechanism, lead to a lack of *m*AST and *m*ALT and, thus, both accumulation of Gln and Glu, as well as reduction in *m*AST/*m*ALT-derived Asp and Ala, which would be offset by anticipated increases due to the loss of cytoplasmic transaminases in hepatocytes (Figure 5).

Notably, the only amino acid for which significantly decreased levels were observed was Cys (Figure 4). In hepatocytes, Cys is essential to GSH biosynthesis and, indeed, the major determinant of GSH availability [45]. Decreased Cys, accompanied by a decrease in GSH, therefore, likely reflects increased utilization of the latter, as part of phase II detoxification, to remove reactive AFB1 epoxides. In light, however, of the apparent effect of AFB1 on both energy and lipid metabolism (as discussed below), depletion of Cys may, alternatively, or additionally, be related to demands for coenzyme A for which Cys is, likewise, an essential biosynthetic building block (Figure 5).

Dysfunction in amino acid metabolism intersects with the observed effects of AFB1 on both lipid and carbohydrate/energy metabolism (Figure 5). Anaplerotic catabolism of amino acids, specifically following deamination, leads to α-ketoacid products, which serve as both intermediates for entry into the Krebs cycle and as substrates for gluconeogenesis, and indirectly, for lipid biosynthesis (i.e., acetyl-CoA), for which the liver is the centrally-functioning organ. Amino acids for which levels were significantly increased by AFB1 exposure include both strictly gluconeogenic (i.e., Glu, Gln, Gly) representatives and essential amino acids (i.e., AAA) that can be either gluconeogenic or ketogenic. In fact, both lipid metabolism and cellular energetics (and related carbohydrate metabolism) have been, likewise, clearly linked to hepatic pathology including AFB1 hepatotoxicity, and all previous metabolomics studies have identified similar alteration of these metabolic pathways [20,21,22,23].

With respect to carbohydrate metabolism and associated cellular energetics, one of the most striking metabolic effects is a significant decrease in G1P, G6P, and Glc accompanied by concomitant increases in Lac and ATP, NADH, and NAD^+^ (Figure 4). Given the unique role of G1P in glycogenolysis (and, in reverse, glycogenesis), increased levels of this intermediate, in concert with decreases in G6P and Glc, are highly suggestive of a breakdown of glycogen to supply glucose for energetic metabolism. 

Decreases in Glc and glycogen-derived intermediates occur alongside an increase in Lac that would be indicative of either anaerobic glycolysis and/or elevated LDH activity. Lactate dehydrogenase (as a stable and ubiquitous cellular enzyme) is, in fact, widely recognized as an indicator of cytotoxicity, in general, and of AFB1 hepatocytotoxicity specifically [29]. Release of this enzyme, following hepatic cell death, would be expected to lead to the production of Lac from pyruvate (derived, in turn, from glycogenolysis and subsequent glycolysis with the attendant increase in ATP and NADH) and a concomitant increase in NAD^+^, as observed (Figure 4 and Figure 5). That said, LDH is also functionally associated with anaerobic glycolysis. Anaerobic glycolysis is, in fact, an essential energetic pathway during early embryonic stages, and a requisite shift in metabolism from anaerobic glycolysis to oxidative phosphorylation accompanies programmed embryo development [46]. As such, the increase in Lac (via anaerobic glycolysis) would be consistent with general impairment of embryo development as previously observed in HRMAS NMR studies of zebrafish exposed to developmental toxins [19]. Either way, shunting of Glc to glycolysis and subsequent consumption of pyruvate (by either LDH release or anaerobic glycolysis) would be consistent with reduced entry into the Krebs cycle (via acetyl CoA). At the same time, the loss of amino acid transaminase activity, in association with hepatotoxicity (as discussed above), would reduce levels of αKG, as a second entry point into the Krebs cycle, which would further compound energetic stress on the developing embryo. 

It is proposed that stage-dependent (and presumably hepatocyte-dependent) toxicity of AFB1, and alterations of metabolic pathways observed here, result from impairment of energy metabolism and associated anaplerotic reactions (e.g., amino acid catabolism, lipid metabolism) during embryo development (Figure 5). The effects of AFB1 on cellular energy, and particularly the Krebs cycle, indeed, have been previously demonstrated in mammalian (i.e., dairy goat) models [22]. Moreover, recent studies have shown that AFB1 targets mitochondria and, specifically, uncouples oxidative phosphorylation [44]. Both disruption of mitochondria, in general, and inhibition of oxidative phosphorylation (as a key developmental transition) could, therefore, explain the impairment of development and general toxicity. Mitochondrial disruption would limit the utility of the Krebs’ cycle, and subsequent oxidative phosphorylation (which is coupled to the Krebs cycle via succinate and NADH), in the mitochondrial matrix, and shift energetic demand to cytosolic glycolysis (leading to the production of Lac). This is further supported by increased acetate, as a proxy for acetyl CoA (as the primary substrate for the Krebs cycle), which may suggest a build-up of this intermediate following disruption of mitochondria (and loss of energetic functionality in the Krebs cycle and oxidative phosphorylation). It has, in fact, been shown that inhibition of the transition from aerobic glycolysis to oxidative phosphorylation during embryo development leads to apoptotic cell-death in progenitor cells [46], which would, therefore, explain both acute toxicity (i.e., lethality) and developmental defects (Figure 1 and Figure 2).

Alongside carbohydrate/energy metabolism, hepatocytes are the primary location of lipid metabolism including both biosynthetic and catabolic functions. Indeed, hepatic damage is clinically associated with lipid accumulation (i.e., hepatic steatosis) or so-called “fatty liver.” Hepatotoxicity of AFB1 would, therefore, also closely correlate with loss of lipid metabolic function and, thus, observed increases in FA and Chol (Figure 4). Indeed, numerous studies (in other model systems) have, likewise, reported increases in lipids (both FA and Chol) in association with AFB1 exposure and toxicity [21,47]. Although acetyl CoA (as the catabolic product of β-oxidation of FA and substrate for lipid biosynthesis) was not directly measured in the present study, acetate may serve as a proxy for impairment of lipid metabolism. The observed increase in acetate may, in this regard, indicates catabolic breakdown, as well as subsequent accumulation of this substrate of the Krebs cycle (due to loss of mitochondrial energy production) and simultaneously as a building block of lipid biosynthesis (aligned with observed increases in FA and Chol). At the same time, the liver produces bile acids, which are primarily derived from cholesterol, and increases in Chol may consequently reflect, in addition, a loss of this function (due to hepatotoxicity). Similarly, alongside disruption of lipid metabolism, observed elevation of lipids (including both FA and Chol) in the present study may be additionally augmented by release from hepatic membranes (i.e., swelling, release, and hydrolysis of phospholipids) in association with hepatic damage (Figure 5; discussed above).

### 3.3. Alteration of Metabolic Profiles in Relation to Neurotoxicity of AFB1

Finally, in addition to recognized hepatotoxicity, AFB1 has been very recently found to impair locomotor function and disrupt neural development in zebrafish embryos and larvae [13]. Although no such neurobehavioral or neural development effects were directly observed or measured in the present study, the significant increases in GABA, Glu, and Gly (Figure 4) would support such effects. Indeed, the developing zebrafish embryo is enriched in several metabolites associated with the CNS including neurotransmitters (such as Glu, Gly, and GABA), likely due to their role in development including, in particular, the neural crest as a key population of progenitor cells. Thus, alongside their roles in neuronal function (as neurotransmitters), all three of these amino acids have been found to have a role in the development of the CNS during embryogenesis [48,49,50], and alterations of their levels may indicate a contribution to the neurobehavioral and neurodevelopmental effects of AFB1 in zebrafish. 

Of these, Glu and GABA are particularly notable given their shared pathways of “recycling” (between neurons and glial cells) via the glutamate/GABA-glutamine cycle and synchronized association with the Krebs cycle (Figure 6). With respect to glutamatergic neurons, synaptic Glu is taken up (via excitatory amino acid transporters) to astrocytes where it is converted to Gln by glutamine synthetase (GS), and subsequently transported to neuronal cells where mitochondrial phosphate-activated glutaminase recycles Gln to the neurotransmitter Glu. In GABAergic neurons, GABA is, likewise, recovered from synapses (though to a lesser extent compared to Glu) by astrocytes and similarly converted to Gln. However, in this case, conversion to Gln occurs by transamination (i.e., GABA transaminase) of α-keto acids (i.e., α-KG, pyruvate or glyoxylate) to amino acids (i.e., Glu, Gly and Ala) and involvement of the Krebs cycle (via succinate semialdehyde, succinate, and αKG, sequentially; Figure 6) to produce Glu and, in turn, Gln (via GS), which is transported to neurons where PAG converts Gln back to Glu and subsequently GABA (via glutamic acid decarboxylase (GAD)). In both cases, Glu can be shunted to the mitochondria to meet energy demands. Thus, the proposed impairment of mitochondrial energy production (i.e., Krebs cycle and subsequent oxidative phosphorylation, as discussed above) would, likewise, lead to increased levels of Glu, GABA, and Gln in neurons. Biosynthesis of Gly, on the other hand, is by way of 3-phosphoglycerate and, subsequently, following transamination by Glu, serine. As the former is a key intermediate in glycolysis, elevated levels of Gly may reflect increased diversion of Glc to this energetic pathway (Figure 5). Regardless of the source, impaired homeostasis of the three neurotransmitters could, in turn, explain possible dysfunction in both neural function and development observed in zebrafish embryos [13]. Based on these findings, a proposed model for AFB1-induced neurotoxicity is summarized in Figure 6.

## 4. Conclusions

Coupling of NMR-based techniques to early life stages of the zebrafish was demonstrated in the present study (and, indeed, other recent studies [15,16,17,18,19]) to provide unique access into the integrated metabolome of an intact organism. Herein, we specifically demonstrated that this approach (when coupled to this established toxicological system) not only revealed a remarkable level of consistency with previous ex vivo metabolomics studies in mammals [20,21,22,23], but simultaneously enabled a holistic model (Figure 5), with respect to the hepatotoxicity of AFB1. As such, this underscores the potential of the technique toward otherwise inaccessible insight regarding pathways and biomarkers of toxicity. As a further demonstration of this potential, alterations of metabolites associated with function and development of the CNS (i.e., neurotransmitters) additionally revealed previously-unknown biochemical effects on cellular homeostasis, which may explain the previously-proposed neurotoxicity of AFB1, including recently-reported neurobehavioral and neurodevelopmental impairment in the zebrafish model [13,51].

## 5. Materials and Methods

### 5.1. Chemicals

All chemicals, including AFB1, were obtained from Sigma-Aldrich (St. Louis, MO, USA), unless otherwise mentioned

### 5.2. Zebrafish Embryos

Adult wild-type zebrafish (*Danio rerio*) were maintained in recirculating aquarium systems according to established rearing procedures [16], and breeding and embryo collection were performed by following the standard procedure as described earlier [16]. Husbandry and experimental procedures (i.e., exposures, collection of embryos) involving zebrafish embryos were performed in accordance with the local animal welfare regulations and maintained according to standard protocols [52]. This local regulation serves as the implementation of the guidelines on the protection of experimental animals by the Council of Europe, Directive 86/609/EEC, which allows zebrafish embryos to be used up to the moment of free-living (5 days after fertilization). Since embryos used in this study were no more than 5 days old, no license is required by the European Union, Directive 2010/63/EU (1 January 2010), or the Leiden University ethics committee. 

### 5.3. Zebrafish Embryo Toxicity Assays

To evaluate acute toxicity and developmental defects caused by AFB1, zebrafish embryos at representative stages (24, 28, 72, and 96 hpf) were treated with varying concentrations of AFB1 (0, 0.25, 0.5, 1.0, and 2.0 µM) for 24 h in 35 mm-diameter polystyrene dishes (*N* = 20 embryos per replicates, i.e., dish, and *N* = 6 replicates per treatment group). Percentage survival of the embryos was evaluated and scored for lethal or teratogenic effects, using a Zeiss CKX41 inverted microscope with phase contrast optics, amounted time-lapse recorder, and the analysis software (Olympus, Hamburg, Germany). Lethal or teratogenic effects were recorded according to Weigt et al. [11]. Teratogenic effects were considered valid if the following criteria were fulfilled:(i) concentration-response relationship and (ii) the endpoint observed in ≥50% of embryos showed teratogenic effects in all replicates. Lethal concentrations for 50% (LC_50_) were calculated by probit analysis. 

### 5.4. HRMAS NMR

Metabolic profiling by HRMAS NMR was performed as adapted from previous studies [18,19]. Embryos (72 hpf) were exposed to 1 µM AFB1 for 24 h; control embryos were exposed to the solvent (i.e., DMSO) carrier only. As the exposure concentration for AFB1 was approximately equal to the LC_50_, additional exposure replicates were done in order to generate a sufficient number of embryos (*N* = 100) and replicates (*N* = 3) for quantitative NMR analyses. Accordingly, ~100 embryos were collected (after 24 h) from both controls (*N* = 3) and pooled (*N* > 3) AFB1 exposures. Following washing (3-times with MilliQ water) to remove residual AFB1, embryos were transferred to 4-mm zirconium oxide rotors (Bruker BioSpin AG, Switzerland) for replicate (*N* = 3) measurements by NMR. As a reference (^1^H chemical shift at 0 ppm), deuterated phosphate buffer (10 µL of 100 mM, pH 7.0) containing 0.1% (*w*/*v*) 3-trimetylsilyl-2,2,3,3-tetradeuteropropionic acid (TSP) was added, and the rotor was transferred immediately to the NMR spectrometer. All HRMAS NMR experiments were done on a Bruker DMX 600-MHz NMR magnet with a proton resonance frequency of 600 MHz, which was equipped with a 4-mm HRMAS dual ^1^H/^13^C inverse probe with a magic angle gradient and spinning rate of 6 kHz. Measurements were carried out at a temperature of 277 K using a Bruker BVT3000 control unit. Acquisition and processing of date were done with Bruker TOPSPIN software (Bruker Analytische Messtechnik, Germany). 

A rotor synchronized Carr–Purcell–Meiboom–Gill (CPMG) pulse sequence with water suppression was used for one-dimensional ^1^H HR-MAS NMR spectra. Each one-dimensional spectrum was acquired applying a spectral width of 8000 Hz, domain data points of 16k, a number of averages of 512 with 8 dummy scans, a constant receiver gain of 2048, an acquisition time of 2 s, and a relaxation delay of 2 s. The relaxation delay was set to a small value to remove short *T_2_* components due to the presence of lipids in intact embryo samples. All spectra were processed by an exponential window function corresponding to a line broadening of 1 Hz and zero-filled before Fourier transformation. NMR spectra were phased manually and automatically baseline corrected using TOPSPIN 2.1 (Bruker Analytische Messtechnik, Germany). The total analysis time (including sample preparation, optimization of NMR parameters, and data acquisition) of ^1^H-HRMAS NMR spectroscopy for each sample was approximately 20 min.

Two-dimensional (2D) homo-nuclear correlation spectroscopy (¹H-¹H COSY), specifically in magnitude mode, was performed using a standard pulse program library. For COSY, 2048 data points were collected in the *t_2_* domain over the spectral width of 4k, and 512 *t_1_* increments were collected with 16 transients, a relaxation delay of 2 s, an acquisition time of 116 ms, and a pre-saturated water resonance during relaxation delay. Data were zero-filled with 2048 data points and weighted with a sine bell window function in both dimensions (prior to Fourier transformation). To preclude the possibility of sample degradation during COSY experiments, the 1D ^1^H HRMAS spectra were measured before and after ^1^H-^1^H COSY measurements (Appendix A). 

### 5.5. ^1^H NMR Data Analysis

All of the spectra were referenced, baseline- and phase-corrected, and analyzed by using MestReNova v.8.0 (Mestrelab research S.L., Santiago de Compostela, Spain). Quantification of metabolites was performed by Chenomx NMR Suite 8.2 (Chenomx Inc., Edmonton, Alberta, Canada). This enabled qualitative and quantitative analysis of an NMR spectrum by fitting spectral signatures from an HMDB database to the spectrum. The concentrations of metabolites were subsequently calculated based on a ratio relative to tCr, since the external reference may lead to the misleading results, and Cr resonance has been previously shown to be a reliable internal reference in a wide range of animal studies. Statistical analysis of NMR quantification was done by one-way analysis of variance (ANOVA) using OriginPro v. 8 (OriginLab, Northampton, MA, USA), and calculated *F*-values larger than 2.8 (*p* < 0.05) were considered significant.

For multivariate analysis, AMIX (Version 3.8.7, BrukerBioSpin, The Woodland, TX, USA) was used to generate bucket tables from the one-dimensional spectra of control and AFB1treated embryos, excluding the region between 4.20 and 6.00 ppm to remove the larger water signal. The one-dimensional CPMG spectra were normalized to the total intensity and binned into buckets of 0.04 ppm. The data were mean-centered and scaled using the Pareto method in the SIMCA software package (Version 14.0, Umetrics, Umeå, Sweden). Unsupervised principle component analysis (PCA) was performed on the data using the SIMCA software as described earlier [19]. 

## Figures and Tables

**Figure 1 toxins-11-00258-f001:**
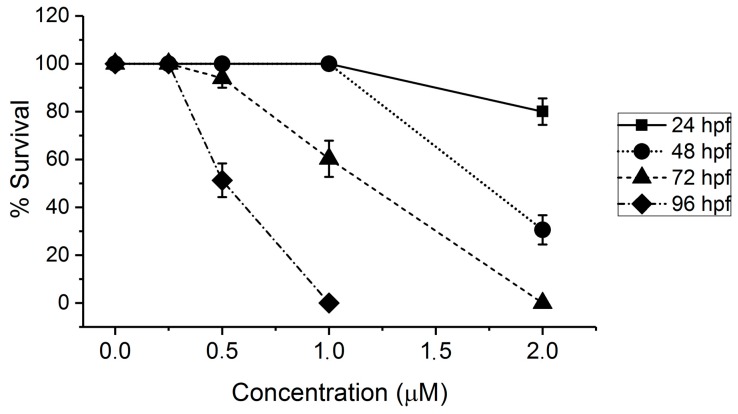
Concentration-dependent toxicity of aflatoxin B1 (AFB1) to zebrafish embryos as measured by lethality. Embryos at 4, 24, 72, and 96 hours post-fertilization (hpf) were exposed (*N* = 6 replicates, *N* = 20 embryos per replicate) to a range of concentrations of AFB1 (i.e., 0.25, 0.5, 1.0, and 2 µM in DMSO) for 24 h. Percentage of survival of the embryos was recorded after 24 h of treatment.

**Figure 2 toxins-11-00258-f002:**
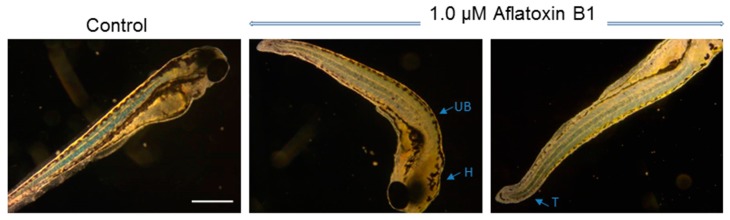
Developmental deformities of zebrafish embryos exposed at 72 h post-fertilization (hpf) to 1.0 µM aflatoxin B1 for 24 h (compared to solvent, i.e., DMSO, only (“control”)). Images were taken at 96 hpf. Deformities include malformation of head (H) and bending of upper body (UB) and tail (T). The scale bar given represents 500 µm (10× magnification).

**Figure 3 toxins-11-00258-f003:**
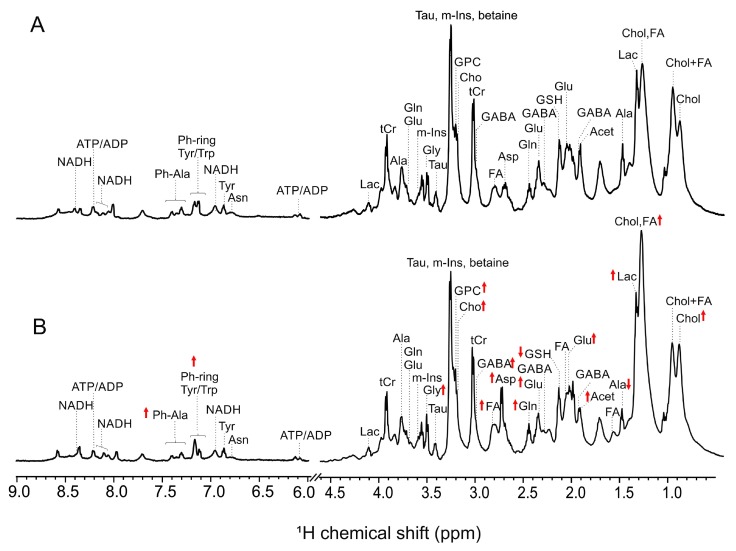
Representative high-resolution magic angle spin (HRMAS) NMR spectra of (**A**) control (i.e., DMSO-only), and (**B**) AFB1-exposed (1 µM) zebrafish embryos exposed at 72 hpf for 24 h. Integrated peak areas and chemical shifts of 1D NMR spectra were used to quantify and identify metabolites. Red arrows indicate increase (↑) and decrease (↓) of metabolites.

**Figure 4 toxins-11-00258-f004:**
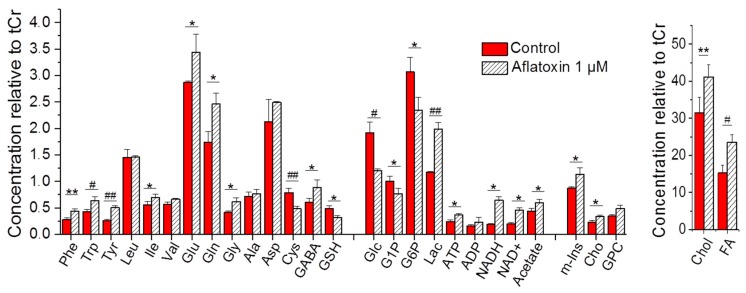
Effect of AFB1 treatment on the metabolic profile of intact zebrafish embryos. Zebrafish embryos (3 dpf) were exposed to 1 µM AFB1 (“aflatoxin 1 µM”) or solvent vehicle, i.e., DMSO, only (“control), for 24 h. Shown are concentrations of metabolites relative to total creatine (tCr); values are the average ± the SE of the mean. Statistical analysis (*t*-test and ANOVAs) of the NMR quantification results were performed with OriginPro v. 8 (Northampton, MA, USA). ## *p* < 0.0001, # *p* < 0.001, ** *p* < 0.01, and * *p* < 0.05. Abbreviations: Phe = phenylalanine; Trp = tryptophan; Tyr = tyrosine; Leu = leucine, Ile = isoleucine; Val = valine; Glu = glutamate; Gln = glutamine; Gly = glycine; Ala = alanine; Asp = aspartate; Cys = cysteine; GABA = γ-aminobutyric acid; GSH = glutathione; Glc = glucose; G1P = glucose-1-phosphate; G6P = glucose-6-phosphate; Lac = lactate; ATP = adenosine triphosphate; ADP = adenosine diphosphate; NADH/NAD^+^ = reduced/oxidized nicotinamide adenine dinucleotide; m-Ins = *myo*-inositol; Cho = choline; GPC = glycerophosphocholine; Chol = cholesterol; FA = fatty acids.

**Figure 5 toxins-11-00258-f005:**
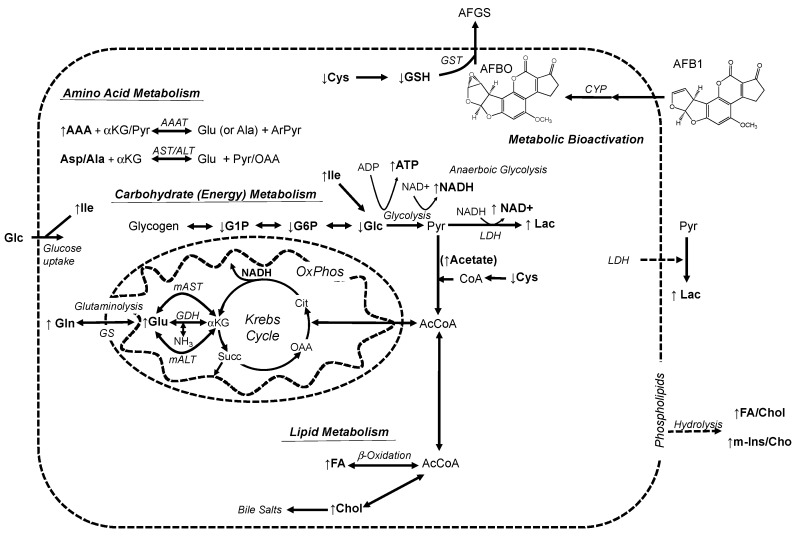
Proposed model of the hepatotoxicity of AFB1 in relation to observed alterations of metabolic profiles. A significant increase and decrease of metabolites, measured in the present study, are indicated by arrows (i.e., ↑ and ↓, respectively). Metabolic bioactivation of AFB1 to the reactive epoxide, AFBO, by cytochrome P450 enzymes (CYP) leads to damage of hepatocytes including cell membrane disintegration and consequent hydrolysis and release of lipids (FA and Chol), as well as polar headgroups (m-Ins and Cho) of phospholipids. Conjugation of GSH by glutathione-S-transferase (GST), as part of phase II detoxification, facilitates removal of AFBO as GSH-conjugate (AFGS), resulting in significant decreases of GSH and its biosynthetic precursor, Cys. Dashed lines indicate loss of cellular (i.e., hepatocyte) and subcellular (i.e., mitochondrial) integrity and function; accordingly, targeting of AFB1 to the liver and consequent hepatocytotoxicity, along with disruption of mitochondria and consequent functional loss of the Krebs cycle, as well as uncoupling of oxidative phosphorylation, would result in altered amino acid, carbohydrate (energy), and lipid metabolism. For the full discussion of the proposed model, relative to alterations in metabolites, see the text. See Figure 4 and the text (Results section), for the abbreviations of measured metabolites. Additional abbreviations: AAA = aromatic amino acids; AAAT = aromatic amino acid transaminases; ArPyr = aromatic pyruvate derivatives; αKG = α-ketoglutarate; (*m*)AST = (mitochondrial)aspartate transaminase; (*m*)ALT = (mitochondrial)alanine transaminase; Pyr = pyruvate; OAA = oxaloacetate; LDH = lactate dehydrogenase; GS = glutamine synthetase; CoA = coenzyme A; AcCoA = acetyl coenzyme A; GDH = glutamate dehydrogenase; Succ = succinate; Cit = citrate; OxPhos = oxidative phosphorylation.

**Figure 6 toxins-11-00258-f006:**
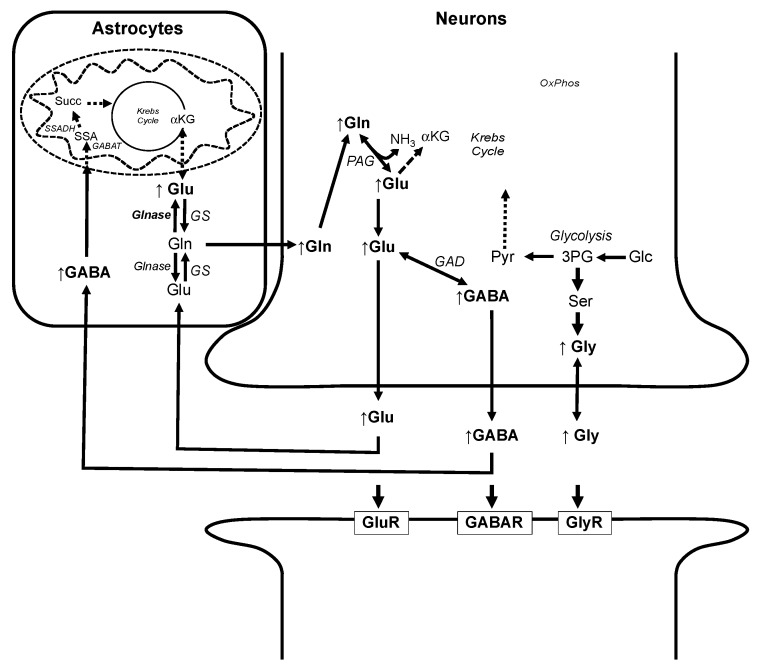
Proposed model of alterations of metabolic profiles in relation to previously-reported effects of AFB1 on neurobehavior and neurodevelopment in zebrafish embryos and larvae [13]. Resultant increases in metabolites (i.e., Gln, Glu, GABA, and Gly) measured in the present study indicated by arrows (**↑**). Mitochondrial disruption, and consequent functional loss of the Krebs cycle, and uncoupling of oxidative phosphorylation [13] (as shown by dashed lines), following AFB1 exposure, results in disrupted homeostasis of neurotransmitters Glu and GABA, as well as Gln as key intermediates in the postsynaptic recycling (i.e., Gln-Glu/GABA cycle) of these neurotransmitters. At the same time, diversion of glycolysis intermediate (i.e., 3-phosphosphoglycerate) to Gly increases this neurotransmitter. For abbreviations, see Figure 4 and Figure 5. Additional abbreviations: GABAT = GABA transaminase; SSA = succinate semialdehyde; SSADH = succinate semialdehyde dehydrogenase; GAD = glutamic acid decarboxylase; PAG = phosphate activated glutaminase; Glnase = cytosolic glutaminase; Ser = serine; 3PG = 3-phosphoglycerate; GluR = glutamate receptors; GABAR = GABA receptors; GlyR = glycine receptors.

**Table 1 toxins-11-00258-t001:** Relative (i.e., percent) change in metabolites and ratio (i.e., fold-change) of aromatic amino acids (AAA) to branched-chain amino acids (BCAA) of zebrafish embryos exposed to AFB1 compared to controls. Embryos exposed to 1 µM AFB1 at 72 hpf (for 24 h) and the concentration of metabolites (relative to total Cr) measured by HRMAS NMR compared to vehicle (DMSO) only controls. For statistically-significant changes, *p*-values are given; “n.s.” indicates that differences are not significant.

Metabolite ^1^	% Change ^2^	*p*-Value			FC ^3^ (T/C)	*p*-Value
**Amino Acids**				**AAA:BCAA**	+1.5	<0.01
Phe	+58.7	<0.01		**Tyr:BCAA**	+1.8	<0.01
Trp	+48.3	<0.001				
Tyr	+97.8	<0.0001				
Leu	+0.4	n.s.				
Ile	+26.8	<0.05				
Val	+16.4	n.s.				
Glu	+20.0	<0.05				
Gln	+41.9	<0.05				
Gly	+49.3	<0.05				
Ala	+8.3	n.s.				
Asp	+17.3	n.s.				
Cys	−38.9	<0.0001				
GABA ^4^	+45.8	<0.05				
**Carbohydrates and Energy Metabolism**						
Glc	−37.2	<0.001				
G1P	−23.6	<0.01				
G6P	−23.6	<0.01				
Lac	+70.4	<0.0001				
Acetate	+36.3	<0.05				
ATP	+52.4	<0.05				
ADP	+49.4	n.s.				
NADH	+234.1	<0.05				
NAD+	+132.5	<0.05				
**Lipids (and Polar Head Groups)**						
Chol	+30.7	<0.01				
FA	+54.5	<0.001				
m-Ins	+29.1	<0.05				
Cho	+48.6	<0.05				
GPC	+42.2	n.s.				
**Glutathione**	−34.8	<0.05				

^1^ For abbreviations, see Figure 4’s legend. ^2^ Percent increase (+) or decrease (−) for metabolite (x) calculated based on concentrations ([]), relative to total Cr, for treated (T, 1 µM AFB1) compared to control (C, DMSO only) as follows: % Change = ([x]_T_ – [x]_C_)/[x]_C_. ^3^ Fold-change (FC) calculated as concentration, relative to total Cr, of total AAA (i.e., Phe, Trp, and Tyr) or Tyr only, relative to the concentration of BCAA (i.e., Leu, Ile, and Val), for treated (T, 1 µM AFB1) divided by control (C, DMSO-only) as follows: FC_AAA/BCAA_ = ([AAA]/[BCAA])_T_/([AAA]/[BCAA])_C,_ and FC_Tyr/BCAA_ = ([Tyr]/[BCAA])_T_/([Tyr]/[BCAA])_C._
^4^ GABA is an amino acid neurotransmitter and not a proteinogenic amino acid.

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
