# Peer review of "NMR-Based Metabolic Profiles of Intact Zebrafish Embryos Exposed to Aflatoxin B1 Recapitulates Hepatotoxicity and Supports Possible Neurotoxicity"

_toxins, 2019, doi:10.3390/toxins11050258_

Reviewer 1 Report

The research presented in this manuscript is interesting and useful not only scientifically, but also in economic and industrial terms. They point to the problem of dangerous fungal infections of grains and other agricultural crops. Fungal diseases can not only affect its productivity but also pollute the grains and related products by generating toxic substances, such as aflatoxin B1 (AFB1) . Fungal infections are a huge problem and a challenge for modern science and food economy. 

The research method was properly selected because the zebrafish model (Danio rerio) is widely used and the most effective assessment system for the detection of environmental toxins. Importantly, the results of a zebrafish toxicity study can be given as an important reference to human beings.

Additionaly, in the presented research, authors used nuclear magnetic resonance (NMR), and specifically high-resolution magic angle spin (HRMAS) NMR, coupled to the zebrafish embryo toxicological model, to characterize metabolic profiles associated with exposure to AFB1.

The results reinforce not only informations about toxicological pathways of AFB,  but also multiple metabolites as potential biomarkers of exposure and toxicity. Furthemore,  these experiments are a powerful toxicometabolomics tool, because coupling of NMR-based techniques to early life stages of the zebrafish was provided unique access into the integrated metabolome of organism. 

 The results are presented in a clear and understandable way and 

To sum up, the study is very interesting and good quality and I recommend this manuscript to the next stages of the editorial process. 

Author Response

The authors appreciate the reviewers positive comments regarding the study, and the manuscript.

Reviewer 2 Report

This is a review of “NMR-Based Metabolic Profiles of Intact Zebrafish Embryos Exposed to Aflatoxin…” The essence of this study is that the author(s) basically demonstrate that metabolism in zebrafish embryos is altered after aflatoxin exposure; this alteration includes higher levels of particular amino acids such as GLU and GLN, and lower levels of other amino acids such as cysteine. The changes in amino acid concentrations also correlated with higher levels of the neurotransmitter GABA.  The authors also show that there are developmental abnormalities in the zebrafish exposed after 72 hrs and that the toxicity of aflatoxin increases with the age of the embryo. The presumption is that phase I (P450 activatin) precedes or depletes phase II (glutathione conjugation) detoxification leading to the time-dependent toxicity.  Overall, both the model organism and the observations are very interesting. The primary message is that AFB1 exposure disrupts the Krebs cycle, leading to more amino acids, such as glutamine, while decreasing levels of glutathione and thus decreasing the level of cysteine. However, the paper can be improved and the authors would do well to pinpoint some of the limitations of their study; namely that the NMR is performed on a group of embryos while the toxicity and deformity assays are measured per single organism.

Major points:

It is important that the authors indicate the number of embryos examined. Figure 1 refers to a percentage, while it is unclear which proportion of embryos exhibit malformations. In Materials and Methods, it is mentioned that 120 embryos were used. If so, a simple numerical number (N = "X" would suffice in the figure legends.

2. The reading of the results section would be facilitated if the major changes in amino acid levels were summarized. It’s not until page 10 that cysteine levels are discussed, yet knowing that cysteine levels are altered helps the reader understand why alterations of the TCA intermediates.

3. The presumption is that the toxicity results when AFB1 is activated but not adequately detoxified. However, there were no measurements of either the phase I or II enzymes involved in metabolic activation.

3. The authors state that diagnosis of aflatoxicosis is difficult. However, there are cases that have been documented and authors would do their readers a service to note a few, such as:

Krishnamachari et al,   Hepatitis due to aflatoxicosis. An outbreak in Western India

Lancet 1975

Tandon,  et al.  Study of an epidemic of jaundice, presumably due to toxic hepatitis, in Northwest India. 

Gastroenterology 1977.  488-494

Ngindu et al. Outbreak of acute hepatitis caused by aflatoxin poisoning in Kenya

Lancet  1982

Jonathan H Williams, Timothy D Phillips, Pauline E Jolly, Jonathan K Stiles, Curtis M Jolly, Deepak Aggarwal, Human aflatoxicosis in developing countries: a review of toxicology, exposure, potential health consequences, and interventions, The American Journal of Clinical Nutrition, Volume 80, Issue 5, November 2004, Pages 1106–1122, https://doi.org/10.1093/ajcn/80.5.1106

Author Response

We thank the reviewer for both positive comments regarding the study and manuscript, and for constructive comments.  The reviewer’s point regarding the limitations of this particular technique, and specifically that it is bulk analysis of pooled embryos, is well taken, and indeed, this technique does not enable assessment of organ, tissue or cell-specific effects (as in assessments of developmental toxicity).  Notably, organ-specific HRMAS NMR can be done on much larger adults, but it is simply not possible on the small (~1 mm diameter and ~1 µL volume) embryos, and this, of course, was not the focus of the present study.  This limitation was, in fact, addressed in detail in our responses to Reviewer 3.  To clarify, lethality as a measure of toxicity was assessed in “bulk” (i.e., calculation of LC50) for embryos, but deformities were not, and no quantitative relationship between concentration and frequency of deformities was observed.  This point was clarified based on the reviewer’s comments in the revised manuscript.

1.  “It is important that the authors indicate the number of embryos examined. Figure 1 refers to a percentage, while it is unclear which proportion of embryos exhibit malformations. In Materials and Methods, it is mentioned that 120 embryos were used. If so, a simple numerical number (N = "X" would suffice in the figure legends.”

The number of replicates (N = 6), and number of embryos per replicate (n = 20 embryos per replicate), in the toxicity assays has been added to the legend for Figure 1.  With respect to frequency of deformities, although no concentration-dependent relationship was measured, the frequency of deformities was approximately 20-30% among surviving embryos (at “sub-lethal” concentrations).  This information has been added to the revised Results (subsection 2.1, second paragraph).  

2.  “The reading of the results section would be facilitated if the major changes in amino acid levels were summarized. It’s not until page 10 that cysteine levels are discussed, yet knowing that cysteine levels are altered helps the reader understand why alterations of the TCA intermediates.”

An overview of all of the metabolites including amino acids for which significant alterations (i.e., increases/decreases) were measured is summarized in the Resultssection (paragraph 2 of subsection 2.2).  However, we do appreciate the reviewer’s point here.  And accordingly, have added a brief summary of the alterations of amino acids specifically in beginning of this section of the Discussion(paragraph 3 of subsection 3.2; see revised version), to effectively “set-up” subsequent discussion of this point.

3.  “The presumption is that the toxicity results when AFB1 is activated but not adequately detoxified. However, there were no measurements of either the phase I or II enzymes involved in metabolic activation.”

The reviewer is correct.  We did not directly measure phase I and II enzymes in this study.  There are, in fact, numerous phase I (i.e., CYP) isoforms, and also multiple interrelated phase II enzymes (e.g., GST, but also many others), and this could potentially be an entire study in itself.  Rather, we used the metabolite profiles from NMR analysis (which DOES include GSH as one of the key phase II peptides) to derive insight as to the possible role of these detoxification systems.  Future studies will likely focus on teasing-out which enzymes in the zebrafish may specifically be involved in the toxicity of AFB1 (to add to the existing knowledge in this regard).

4.   “The authors state that diagnosis of aflatoxicosis is difficult. However, there are cases that have been documented and authors would do their readers a service to note a few, such as:

Krishnamachari et al,   Hepatitis due to aflatoxicosis. An outbreak in Western India

Lancet 1975

Tandon,  et al.  Study of an epidemic of jaundice, presumably due to toxic hepatitis, in Northwest India. 

Gastroenterology 1977.  488-494

Ngindu et al. Outbreak of acute hepatitis caused by aflatoxin poisoning in Kenya

Lancet  1982

Jonathan H Williams, Timothy D Phillips, Pauline E Jolly, Jonathan K Stiles, Curtis M Jolly, Deepak Aggarwal, Human aflatoxicosis in developing countries: a review of toxicology, exposure, potential health consequences, and interventions, The American Journal of Clinical Nutrition, Volume 80, Issue 5, November 2004, Pages 1106–1122, https://doi.org/10.1093/ajcn/80.5.1106”

We agree with the reviewer, and actually had included some of these details  - including, in particular, references to recorded outbreaks of aflatoxicosis – but subsequently removed them from the submitted version of the Introduction for the sake of conciseness.  However, based on the reviewer’s advice, we have now added this point back to the Introduction, and have specifically added a reference to a comprehensive review on the subject (Williams et al., 2004; as per the reviewer’s recommendation).

Reviewer 3 Report

Referee evaluation of the manuscript “NMR-Based Metabolic Profiles of Intact Zebrafish Embryos Exposed to Aflatoxin B1 Recapitulates Hepatotoxicity, and Supports Possible Neurotoxicity” (Ref. no.: TOXINS-489507)

The manuscript is a novel approach, since it tests NMR technique in the measurement / quantification of multiple biochemical compounds, and their mycotoxin induced alterations in Zebrafish, as a model system.

Basically, the approach is acceptable. The parallel measurement of 19 compounds in a very tiny model animal is very interesting. Morever, the possible models evolved for the mode of action are also clearly understandable and are mostly satisfactory.

Thus, this is a well-conceived study. I have only some, but mostly rather important comments.

Major questions:

1. The technical parameters for the NMR measurement are based on the measurement of 3 individuals harvested from the same group, in parallel. According to the tiny size, this is acceptable. BUT: the NMR decay is thus gained from a likewise whole body approach. Indeed, no problem. BUT: whole body analysis has some major points, that preclude some statements.

These are:

LDH and Lac, L236: the results on whole body Lac and LDH are acceptable, but citotoxic effect (cell membrane damage) is not depending on Lac and LDH, but on their intra- and extracellular presence. This must be corrected.

The same is valid for L273, on amino acids.

2. It is somehow unclear in the whole work, how individual amino acids were quantified and why those are handled as biomarkers. NMR in this means is a whole body approach and AA presence, whether present in proteins, oligopeptids or as a free acid is unknown. This must be considered by all AA-s.

3. Cellular level, and more precisely, subcellular level alterations can not be analysed this way - please re-consider “hepatocellular” term at L300-301. Not even the liver can be separately identified in a whole body approach. The same is valid for “plasma” AST and ALT at L313.

4. L287-292: The explanation on ILE is not clear. If ILE is increasing, where? The yolk is already incorporated. Authors can not distinguish between abdominal yolk and other body region in a whole body case. Thus, somehow, rather the absortion rate of ILE across the CAM must be influenced by AFB1. Not?

Minor points:

1. I would not recommend to introduce the technical background in a case where toxicology is the primary goal. (L90-101)

2. Figure 1 has printing problem on the X axis, by micromol.

3. Please add a scale bar to Figure 2.

4. Please increase compound name readability in Figure 3.

5. L146 and for alpha-keto-glutarate at many places: please check the special characters.

6. L150: some information is present here that would rather fit into the discussion.

7. L153: please clarify, what is meant under FA? NEFA, esterified FA, albumin bound form ...? Again, I would not deeply discuss this compound, if its biochemical form is not fully elucidated.

8. Table 1. Lac, ATP and further compounds are not carbohydrates!! Please check. Moreover, choline is not a lipid.

9. Glutathione: this compound can be present in reduced and oxidized form. Detailed explanation without specific information on its redox state is problematic.

Summarized:

The manuscript is generally very well conceived. Its results are novel and correct. Anyway, dataset evaluation can be effectively upgraded to a much more developed form.

I suggest the modification of the ms. taking all my critical points into consideration.

/* Layout-provided Styles */ div.standard { margin-bottom: 2ex; }

Author Response

Major questions:

1. “The technical parameters for the NMR measurement are based on the measurement of 3 individuals harvested from the same group, in parallel. According to the tiny size, this is acceptable. BUT: the NMR decay is thus gained from a likewise whole body approach. Indeed, no problem. BUT: whole body analysis has some major points, that preclude some statements.

These are:

LDH and Lac, L236: the results on whole body Lac and LDH are acceptable, but citotoxic effect (cell membrane damage) is not depending on Lac and LDH, but on their intra- and extracellular presence. This must be corrected.

The same is valid for L273, on amino acids.”

The reviewer is correct that tissue-specific and cellular distribution of metabolites can’t be resolved by the approach used, ostensibly representing a “limitation” of this particular technique. Such measurements for embryos are simply not possible by current NMR capabilities due, in a large part, to the small size (~ 1 mm diameter, ~1 microliter volume) of embryos.  Notably, however, such measurements are, in fact, possible for much larger adult zebrafish, but that was not the focus here.  That said, the NMR analysis which was done was perhaps not clear: in this case, the technique used was a bulksample analysis (effectively pooling all cells and tissues within embryo).  And moreover, we did NOT measure 3 embryos, but rather 3 replicatesof 100 pooled embryos.  This latter point was explained in the Methods and Materials, but we have revised this section to perhaps make more clear (see revised version).

Although measuring tissue and cellular distribution of metabolites in embryos would be ideal, the bulk measurements of overall and pooled changes in metabolites in the whole embryos is, in fact, consistent with the proposed model (see Figure 5) in that all of the alterations (increases/decreases) are generally due to either catabolic or anabolic metabolism (and not release or redistribution within or between cellular, tissue or organ compartments in the embryo, e.g., release to plasma).  While these metabolic changes are primarily considered with respect to “hepatotoxicity” (and thus hepatocytes), this is because of the recognized role of hepatocytes (in bioactivation and subsequent phase II detoxification) in AFB1 toxicity, and moreover, predominant role of the liver in all three metabolic pathways (i.e., amino acids, lipids and carbohydrate/energy metabolism) seemingly affected by the toxin, as well as the observed stage-dependent effects of the toxin.  The only exception is the discussion of Glu/Gln and Gly with respect to possible neurotoxicity since these amino acids do, indeed, alongside their roles as neurotransmitters, have essential roles as proteinogenic amino acids in all cells. However, these changes are discussed here (within the context of neurotoxicity) alongside GABA that is uniquely associated with neurons and neuronal function.

2. “It is somehow unclear in the whole work, how individual amino acids were quantified and why those are handled as biomarkers. NMR in this means is a whole body approach and AA presence, whether present in proteins, oligopeptids or as a free acid is unknown. This must be considered by all AA-s.”

1H-NMR  signals (i.e., chemical shifts, peak areas) used to identify and  quantify amino acids are specific for “free” amino acids, and do not  include those amino acids incorporated into proteins, or other  peptides.  This was clarified in the Resultsof the revised version (see second paragraph of subsection 2.2)

3.  “Cellular level, and more precisely, subcellular level alterations can  not be analysed this way - please re-consider “hepatocellular” term at  L300-301. Not even the liver can be separately identified in a whole  body approach. The same is valid for “plasma” AST and ALT at L313.”

As  per our response to comment #1 (see above), the bulk analyses of  metabolites is consistent with the proposed model  - of hepatocellular  targeting - given that (1) all measured changes in metabolites are due  to either catabolism or anabolism, and not redistribution, i.e.,  release, etc., of metabolites, and  the three metabolic pathways related  to amino acids, lipids and carbohydrates/energy are primarily localized  to the liver; and (2) AFB1 is known to target the liver and  hepatocytes, where bioactivation to epoxides occurs, and this role of  the liver is very consistent with the stage-dependent toxicity which  correlates with development of the liver (and relevant phase I and II  enzymes).

4.  “L287-292:  The explanation on ILE is not clear. If ILE is increasing, where? The  yolk is already incorporated. Authors can not distinguish between  abdominal yolk and other body region in a whole body case. Thus,  somehow, rather the absortion rate of ILE across the CAM must be  influenced by AFB1. Not?”

Again,  we are not proposing any tissue specific distribution of metabolites  (as this is not possible with the bulk NMR analysis of embryos).  As  discussed in the manuscript (see Discussion), the observation  of Ile is notable, however, as it is specifically associated with uptake  and utilization of glucose (and decreased gluconeogenesis) in the  liver.  The other BCAA (i.e., Val and Leu) are not strongly associated  in this way, and are rather associated with non-liver cells (and  particularly skeletal muscle which is presumably not affected by  AFB1).  To address the reviewer’s questions:

1.    The  yolk is NOT completely assimilated at this stage of  development.  Although, 96 hpf embryos are hatched and free swimming,  the yolk is present (as a source of nutrients) until 5-7 dpf.

2.    Ile  is an essential amino acid that, indeed, is derived from yolk (as  pointed-out by the reviewer) - and prior to this from maternal diet -  but subsequently catabolized by either glucogenic or ketogenic pathways  which are both primarily mediated by the liver.  As metabolites present  in yolk (see above points) are included in the bulk analysis, the  observed increase of Ile is presumably not due to uptake of Ile by  embryos (from yolk), but rather reduced catabolism (by liver) of this  amino acid, as per the proposed model of hepatocellular targeting (see  Figure 5, and text of the Discussion).

Minor points:

1.  “I would not recommend to introduce the technical background in a case where toxicology is the primary goal. (L90-101)”

As  per the reviewers suggestion, we have removed some of the technical  explanation regarding HRMAS NMR (i.e., how it works) from the revisedIntroduction.

2.  “Figure 1 has printing problem on the X axis, by micromol.”

Yes,  the x-axis of Figure should be in units of micromolar (µM)  concentration.  It seems the symbol was “lost in translation” during  conversion to PDF. It has been corrected in the current Word document,  and hopefully will be retained in subsequent steps.  We will be sure to  confirm that it is correct in galley proofs.

3.  “Please add a scale bar to Figure 2.”

A scale bar (500 µm) has been added to the figure, and note for this made in the legend for the figure.

4.  “Please increase compound name readability in Figure 3.”

As requested, the size of the labels (i.e., metabolite names) for peaks has been increased in Figure 3.

5.  “L146 and for alpha-keto-glutarate at many places: please check the special characters.”

Yes,  the reviewer is correct: in many places, the alpha symbol was “lost”  presumably in “translation” by font changes.  We would thank the  reviewer for catching this. These have been corrected, and will  hopefully stay fixed; however, we will be sure to confirm this is the  case when we receive galley proofs of the paper.

6.  “L150: some information is present here that would rather fit into the discussion.”

We thank the reviewer for catching this.  This information has been removed, and is, in fact, already included in the Discussionsection.

7.  “L153:  please clarify, what is meant under FA? NEFA, esterified FA, albumin  bound form ...? Again, I would not deeply discuss this compound, if its  biochemical form is not fully elucidated.”

“Fatty acids” ( or “FA”) here represents a non-specific classification based on multiple 1H-NMR  signals (i.e., chemical shifts; see Figure 3) which identify fatty  acids generally and collectively, but does not distinguish particular  subclasses. Although fatty acids can certainly be fully elucidated by  solution-state NMR (in purified form), this is not possible within  complex matrices.  In the context of this paper, the discussion of this  point, therefore, does not elaborate any particular class of fatty  acid. 

8.  “Table 1. Lac, ATP and further compounds are not carbohydrates!! Please check. Moreover, choline is not a lipid.”

The  reviewer is, of course, correct on this point.  To clarify, we grouped  ATP, Lac, etc. with carbohydrate and “energy” metabolism (please refer  to the table) as they are ostensibly part of the same catabolic  pathways; that is to say, that catabolism of glucose produces ATP, NADH,  NAD+ and Lac (to make cellular energy).  We have clarified this  heading/category in the revised Table 1, and it will hopefully make it  clearer.

Similarly,  with respect to choline (as well as myo-inositol and GPC), these were  included with lipids as they are largely found as polar head-groups of  numerous lipids, and particularly phospholipids (i.e.,  phosphatidylcholine and phosphatidylinositol lipids).  And it is argued  (in the manuscript) that hydrolysis, following cell membrane disruption,  releases these head groups from lipids.  To clarify, this delineation  (as “polar head groups”) has been added to the table.

9.  “Glutathione: this compound can be present in reduced and oxidized  form. Detailed explanation without specific information on its redox  state is problematic.”

1H-NMR  signal (in HRMAS NMR) does not distinguish reduced (GSH) and oxidized  (GSSG) glutathione.  However, with respect to phase II detoxification as  proposed, glutathione (as the GSH form) is covalently and irreversibly  conjugated to AFB1 epoxides for removal (e.g., increased water  solubility) such that the pool of both GSH and GSSG would be depleted as  observed here.  Recycling between the two forms (GSH/GSSG) is not  relevant for this mechanism.

Round  2

Reviewer 3 Report

I have 2 minor, merely technical notes:

- fig 1, x axis label is still unchanged

- L217: Methods and Materials - opposite order is the correct version.

Author Response

Thank you to the reviewer for catching these minor issues.  We have now corrected both in the second revision.  

 The symbol (for micro-) in the x-axis of Figure 1 was corrected again in the figure file, and will hopefully be retained in the conversion to PDF.  It does, in fact, appear correctly in the previous Word document, but was somehow corrupted again when converted to PDF.  As mentioned in our previous response, we will keep an eye out when we see the galley proofs to be sure it stays corrected.

"Methods and Materials" was corrected to "Materials and Methods" on line 271.